# Loss of ERβ in Aging LXRαβ Knockout Mice Leads to Colitis

**DOI:** 10.3390/ijms241512461

**Published:** 2023-08-05

**Authors:** Xiaoyu Song, Wanfu Wu, Yubing Dai, Margaret Warner, Ivan Nalvarte, Per Antonson, Mukesh Varshney, Jan-Åke Gustafsson

**Affiliations:** 1Center for Nuclear Receptors and Cell Signaling, Department of Biology and Biochemistry, University of Houston, Houston, TX 77204, USA; xsong7@central.uh.edu (X.S.); wwu16@central.uh.edu (W.W.); daifengdb555@yahoo.com (Y.D.); mwarner@central.uh.edu (M.W.); 2Department of Biosciences and Nutrition, Karolinska Institutet, 14186 Huddinge, Sweden; ivan.nalvarte@ki.se (I.N.); per.antonson@ki.se (P.A.); mukesh.varshney@ki.se (M.V.)

**Keywords:** liver X receptor, estrogen receptor β, colitis, inflammatory bowel diseases, plectin

## Abstract

Liver X receptors (LXRα and LXRβ) are oxysterol-activated nuclear receptors that play key roles in cholesterol homeostasis, the central nervous system, and the immune system. We have previously reported that LXRαβ-deficient mice are more susceptible to dextran sodium sulfate (DSS)-induced colitis than their WT littermates, and that an LXR agonist protects against colitis in mice mainly via the regulation of the immune system in the gut. We now report that both LXRα and LXRβ are expressed in the colonic epithelium and that in aging LXRαβ^−/−^ mice there is a reduction in the intensity of goblet cells, mucin (MUC2), TFF3, and estrogen receptor β (ERβ) levels. The cytoplasmic compartment of the surface epithelial cells was markedly reduced and there was a massive invasion of macrophages in the lamina propria. The expression and localization of β-catenin, α-catenin, and E-cadherin were not changed, but the shrinkage of the cytoplasm led to an appearance of an increase in staining. In the colonic epithelium there was a reduction in the expression of plectin, a hemidesmosome protein whose loss in mice leads to spontaneous colitis, ELOVL1, a fatty acid elongase protein coding gene whose overexpression is found in colorectal cancer, and non-neuronal choline acetyltransferase (ChAT) involved in the regulation of epithelial cell adhesion. We conclude that in aging LXRαβ^−/−^ mice, the phenotype in the colon is due to loss of ERβ expression.

## 1. Introduction

Ulcerative colitis and Crohn’s disease represent the two main types of inflammatory bowel diseases (IBDs). The etiology and underlying molecular mechanisms behind these idiopathic IBDs are currently poorly understood, but involve the activation of mucosal/epithelial cells driving an uncontrolled inflammatory response in the small and large intestines. The intestinal mucosa is in constant contact with microorganisms, and thus the intestinal epithelial barrier is a key component of the defense mechanisms required to prevent infection and inflammation. Intestinal epithelial cells play an important role in controlling the interaction between the luminal microbiota and underlying immune cells [1].

Liver X receptors (LXRs) encoded by the genes *Nhr1h3* (LXRα) and *Nhr1h2* (LXRβ) are members of the nuclear receptor family of ligand-regulated transcription factors [2,3]. LXRs are master regulators of lipid metabolism, cholesterol homeostasis, and inflammation [4,5], and as such, are involved in numerous physiological and pathological processes such as atherosclerosis, cancer, diabetes mellitus, and the loss of specific neurons in the substantia nigra, spinal cord, retina, and cochlea. LXRs are receptors with potent anti-inflammatory effects and have emerged as important targets for inflammatory diseases [6,7]. In addition to synthetic LXRs agonists, such as T0901317 and GW3965, sterol-based LXR agonists attenuate dextran sulfate colitis-induced weight loss with a reduced expression of inflammatory markers in the large intestine of mice [8]. A recent study showed that LXRαβ expression was decreased in colonic samples from patients with IBD and IL-10 deficient mice [9]. In this study, the authors concluded that LXRs exert anti-inflammatory effects in colonic epithelial cells via ABCA1.

It has been reported that genetic variants in the nuclear receptors including LXRs may indicate a risk of ulcerative colitis [10]. In a previous study, we found that either LXRα-, LXRβ-, or LXRαβ-deficient mice were more susceptible to dextran sodium sulfate (DSS)-induced colitis. Clinical indicators of colitis including weight loss, diarrhea, and blood in feces appeared earlier and were more severe in LXRαβ-deficient mice and, particularly, LXRβ protected against symptoms of colitis [11]. Furthermore, the activation of LXRs strongly suppressed the expression of inflammatory mediators. The expression of both LXRα and LXRβ was significantly suppressed in the inflamed colon of IBD subjects compared to the non-inflamed colon [11].

Estrogen and its receptors, in addition to their role in regulating several biological functions, contribute to the development and progression/severity of IBDs [12]. We have previously reported that estrogen receptor β (ERβ) is essential for the maintenance of cellular homeostasis and for driving cellular differentiation in the colon [13]. In ERβ^−/−^ mice, there is rapid proliferation of the epithelium and defective tight junctions characterized by a decrease in the expression of the adherens junction protein, α-catenin, and the hemidesmosomal protein, plectin. ERβ deficiency impairs gut microbiota [14] and intestinal ERβ suppresses colon inflammation and tumorigenesis in both sexes [15].

The aim of this study was to investigate the mechanisms through which loss of LXRs leads to the sensitivity of the colon to colitis. We were particularly interested in the possibility that ERβ signaling was involved because of the colonic phenotype of ERβ mice and our finding of the loss of ERβ in the lungs of LXRαβ^−/−^ mice: aging LXRαβ^−/−^ lungs developed a rare form of lung cancer after mice turned 13 months of age [16]. The hypothesis that we are testing is that loss of ERβ is a key factor in the development of the phenotype in LXRαβ^−/−^ mice because of the loss of the antiproliferative role of ERβ in epithelial cells.

In the present study, we found a colitis phenotype in LXRαβ^−/−^ mice as they age. We examined gene expression via immunohistochemistry and RNA sequencing. We now confirm that LXRβ and LXRα are expressed in colonic epithelial cells and that as LXRαβ^−/−^ mice age (above 12 months), there is dysfunction of the colonic epithelium and colitis, similar to that seen in plectin knockout mice. The study shows that the colonic phenotype in these aged mice is not directly due to loss of LXRs, but was indirectly caused by the loss of ERβ in the colonic epithelium, which occurs in the absence of LXRs.

## 2. Results

### 2.1. Expression of LXRα and LXRβ in the Colon

To detect the expression of LXRs in the colon of WT mice, we used our in-house antibodies which have been well characterized for staining LXRs in the central nervous system and other organs [17,18,19,20,21,22]. With our home-made antibodies, we found that both LXRα and LXRβ are expressed in the colon both in the cells at the base of the crypts, along the crypts, and in the surface epithelium (Figure 1A,B).

### 2.2. Decrease in the Intensity of Goblet Cells and Mucins in the Colon of LXRαβ^−/−^ Mice

To examine the effect of LXRαβ loss on colon morphology, colon sections from LXRαβ^−/−^ and their WT littermates were stained with hematoxylin/eosin. Compared with WT, the colonic epithelium of LXRαβ^−/−^ mice was flattened, with more cells (macrophages) infiltrating the lamina propria (Appendix A). In addition, the number of proliferating cells (PCNA-positive) was higher in the colon of LXRαβ^−/−^ mice and the proliferating cells were not confined to the base of the crypts but migrated up the sides of the crypts (Appendix A).

Goblet cells are found scattered in between enterocytes and secrete high-molecular-weight glycoprotein mucins. Alcian blue and periodic acid–Schiff staining were used to measure the number of goblet cells and mucin content in colonic mucosa. We found that compared to WT mice, the number of goblet cells was significantly reduced in the colon of LXRαβ^−/−^ mice (Figure 2A–F).

Goblet cells produce and secrete mucin 2 (MUC2) along with trefoil factor 3 (TFF3) into the lumen within the intestine cells where they participate in mucosal regeneration and repair [23]. We confirmed a significant reduction in mucin expression using antibodies for MUC2 and TFF3, respectively (Figure 3A–D). Overall, the goblet cell numbers and mucin expression were significantly reduced in the colon of LXRαβ^−/−^ mice.

### 2.3. Increased Activation of Macrophages in the Lamina Propria of LXRαβ^−/−^ Mice

Intestinal macrophages play a key role in the gut immune system and the regulation of gastrointestinal physiology [24]. LXRs are key regulators of the immune system and macrophages [22]. There were a few macrophages (Iba1-positive) detected in the lamina propria of WT mice (Figure 4A,B). However, there were abundant macrophages in the colon of LXRαβ^−/−^ mice (Figure 4C–E). These macrophages appear to be activated, as evidenced by larger cell bodies and limited ramifications. Furthermore, there were only a few MPO-positive neutrophils in the colon of both WT and LXRαβ^−/−^ mice, and there was no difference in the number of neutrophils (Appendix A).

### 2.4. Decreased Expression of Plectin in the Colon of LXRαβ^−/−^ Mice

Plectin in intestinal epithelial cells is important for maintaining keratin filament network organization and the formation of functional hemidesmosomes, as well as intercellular junctions. We observed abundant plectin expression in the colonic epithelium of WT mice, with a marked expression of plectin in the cytoplasmic and perinuclear regions of the colon in WT mice (Figure 5A,B). However, plectin expression was significantly reduced in the colonic epithelium of LXRαβ^−/−^ mice (Figure 5C,D).

### 2.5. Decreased Expression of ChAT and ELOVL1 in the Colonic Epithelium of LXRαβ^−/−^ Mice

Choline acetyltransferase (ChAT) and ELOVL fatty acid elongase 1 (ELOVL1) have been implicated in diseases of the colon. Both enzymes were very abundantly expressed in the colonic epithelium of WT mice, whereas their expression levels were significantly reduced in the colonic epithelium of LXRαβ^−/−^ mice (Figure 6 and Figure 7). In neurons and most non-neuronal cells, ChAT is the enzyme responsible for the synthesis of ACh from choline and acetyl coenzyme A [25] and ELOVL1 catalyzes the elongation of short-chain fatty acids (SCFA).

Adherens junctions (expressing β-catenin, α-catenin, and E-cadherin), together with tight junctions, form an apical junctional complex that regulates intestinal epithelial cell-to-cell adherence and barrier homeostasis [26]. β-catenin is well expressed in the colon of both WT and LXRαβ^−/−^ mice (Appendix A), and the expression of β-catenin in the colonic epithelium of LXRαβ^−/−^ mice appeared to be increased somewhat due to the shrinkage of the epithelium cytoplasm. The expression and localization of α-catenin and E-cadherin in the colon of WT and LXRαβ^−/−^ mice did not significantly change (Appendix A).

### 2.6. ERβ Expression in the Colon of WT and LXRαβ^−/−^ Mice

ERβ was expressed in the colon surface epithelium and cells at the base of the crypts of WT mice (Figure 8A,B). However, ERβ expression was lost in the colonic epithelium of LXRαβ^−/−^ mice (Figure 8C,D).

## 3. Discussion

In the present study, we report that LXRα and LXRβ are expressed in the colonic epithelial cells and that loss of LXRαβ leads to a phenotype of colitis as mice age. This colitis is characterized by loss of goblet cells, increased macrophage activation, and decreased ERβ expression in the colon.

Plectin, a cytoskeletal linker protein, binds to and interconnects intermediate filaments and attaches them to junctional complexes at the cell membrane, different organelles, and inner membrane systems. Plectin also interacts with actomyosin structures and microtubules [27]. It functions as a mediator of keratinocyte mechanical stability in the skin, primarily through linking intermediate filaments to hemidesmosomes [28]. Plectin expression was reduced in the colons of ulcerative colitis patients and negatively correlated with the severity of colitis. Plectin plays an important role in maintaining the integrity of the intestinal epithelium [29]. Mechanistically, plectin deficiency in intestinal epithelial cells leads to aberrant keratin filament network organization and the formation of dysfunctional hemidesmosomes and intercellular junctions. In the present study, we found that the knockout of LXRαβ resulted in significantly reduced plectin expression in the colon. Thus, the colonic phenotype of LXRαβ^−/−^ mice is at least partially mediated by the downregulation of plectin, and effects on the entire integrin α6/plectin/keratin 18 axis are to be expected.

Interestingly, although in the LXRαβ^−/−^ mice there is increased proliferation, there is no significant change in crypt length. This is different from mice bearing an intestinal-epithelial-cell-specific loss of α6 integrin [30], or mice with Keratin 8 deletion [31]. Integrin α6 knockout mice spontaneously develop colitis, which degenerates over time into infiltrating adenocarcinoma [30]. Mutant crypts in the distal colon of α6 integrin knockout mice were twice as long, and proliferation rates in the colon and rectum were significantly increased. Keratin 8 gene disruption led to colorectal hyperplasia with an increased number of gut epithelial cells, and more than five-fold elongated villi [31]. Mice lacking plectin in intestinal epithelial cells spontaneously developed colitis characterized by the extensive detachment of intestinal epithelial cells from the basement membrane and increased intestinal permeability, and exhibited increased crypt proliferation, severe crypt damage, and increased goblet cells [29]. The LXRαβ^−/−^ mouse colon, therefore, most closely resemble plectin knockout mice.

Acetylcholine (ACh) is not only a neurotransmitter but also exists in non-neuronal cholinergic systems. Components of the non-neuronal cholinergic system, including ChAT, are located in intestinal epithelial cells, crypt–villus organoids, immune cells, intestinal stem cells, and vascular endothelial cells [32]. Upon exposure of the SCFA, propionate, which is produced by gut bacteria, ACh is released [33]. ACh is a key regulator of epithelial tightness in the proximal colon [34], and in patients with ulcerative colitis, ChAT expression is reduced in endocrine and epithelial cells of the mucosa [35]. Thus, the non-neuronal cholinergic system plays a significant role in intestinal epithelial homeostasis by protecting the colonic epithelium from changes in the luminal environment [36,37]. In the present study, the expression levels of ChAT in the colonic epithelium of LXRαβ^−/−^ mice were significantly decreased, indicating a disturbance of the non-neuronal cholinergic system. The mechanisms through which LXRs regulate the cholinergic system are not fully understood.

ELOVL1 is involved in the elongation of saturated fatty acids and monounsaturated fatty acids. ELOVL1 is essential for permeability barrier formation, as ELOVL1-deficient mice die shortly after birth due to epidermal barrier defects [38]. LXRs have been implicated in the control of fatty acid synthesis and subsequent lipogenic processes in several tissues.

The intestinal mucosa is in continuous contact with microorganisms, and the intestinal epithelial barrier is a critical component in the defense against infection and inflammation. MUC2, which is produced by the goblet cells, is the major component of the intestinal mucus and protects the intestinal tract from self-digestion and numerous microorganisms [39]. TFF3 is expressed with MUC2 in intestinal goblet cells where it participates in mucosal regeneration and repair [23,40]. In the present study, we observed a significant reduction in the number of goblet cells and decreased expression levels of MUC2 and TFF3 in the colon of LXRαβ^−/−^ mice, suggesting a compromised intestinal barrier in LXRαβ^−/−^ mice.

One possible explanation for the changes in the colonic mucosa as mice age is the rapid turnover of the surface epithelium with an overall consequence on a less-well-differentiated surface epithelium. Late-onset phenotype characteristics of LXR^−/−^ mice have been reported in dopaminergic neurons in the substantia nigra [41], large motor neurons in the ventral horn of the spinal cord [42], retinal ganglion cells in the eye [21], spiral ganglion neurons in the cochlea [22], and squamous cell lung cancer, a rare subtype of non-small cell lung cancer, in LXRαβ^−/−^ mice [16,43]. In the case of non-small cell lung cancer, there was a loss of ERβ expression which contributed to increased proliferation.

ERβ regulation of inflammation and its protective role in colitis and colitis-associated colorectal cancer (CRC) have been extensively studied in terms of other different mechanisms [44]. 17β-Estradiol induces the expression of MUC2 and tight junction molecules and inhibits pro-inflammatory cytokines to exert anti-inflammatory effects on the intestinal barrier through the ERβ signaling pathway [45]. Arctigenin, a phytoestrogen, can maintain the integrity of the mucus barrier by inhibiting the apoptosis of goblet cells via ERβ signaling [46]. Orally administered estrogen receptor modulator (SERM2) could alleviate DSS-induced colitis in male mice and promote anti-inflammatory phenotypes of immune cells [47]. However, the possible regulation of ERβ on plectin, ELOVL1, and ChAT in colitis still needs more animal model experiments and clinical studies to investigate.

One other factor contributing to the immaturity of the colonic surface epithelium is the loss of ERβ expression as LXRαβ^−/−^ mice age. We have previously shown a role for ERβ in the organization and architectural maintenance of the colon, and loss of ERβ leads to colitis in mice. The ERβ^−/−^ mouse colon was characterized by increased proliferation, an increase in the migration of proliferating cells toward the luminal surface, a decrease in apoptosis, and poorer differentiation of the surface epithelium, as measured via the expression of cytokeratin 20, α-catenin, and plectin [13]. Tight junction defects were confirmed via electron microscopy. Furthermore, our results indicate that rapidly proliferating colonic epithelial cells in ERβ^−/−^ mice are lost by increased shedding and not by increased apoptosis. The late development of colitis in LXRαβ^−/−^ mice coincided with the loss of ERβ.

## 4. Materials and Methods

### 4.1. Animals

In this study, 12- and 18-month-old (*n* = 3, *n* = 2) WT mice and 12- and 15-month-old (*n* = 3, *n* = 2) LXRαβ^−/−^ mice were used for histological assays. LXRαβ^−/−^ mice with the C57BL/6 genetic background have been described [20,43,48]. Mice were housed in a room of standard temperature (22 ± 1 °C) with a regular 12 h light, 12 h dark cycle and given free access to water and standard rodent chow. All animal experiments were carried out according to the National Research Council’s Guide for the Care and Use of Laboratory Animals and in accordance with the University of Houston guidelines on the use of laboratory animals. Effort was made to minimize the number of animals used and their suffering. All mice were anesthetized deeply with CO_2_ and transcardially perfused with 0.1 M PBS followed by 4% paraformaldehyde. Colon tissues were dissected and post-fixed overnight at 4 °C using the same fixative. After fixation, colons were processed for paraffin sections.

### 4.2. Histopathological Analysis

Swiss-roll colon sections (5 μm thick) were processed for hematoxylin/eosin (Sigma-Aldrich, St. Louis, MO, USA), periodic acid–Schiff (Chromaview, Kalamazoo, MI, USA), and Alcian blue (Electron Microscopy Sciences, Hatfield, PA, USA) staining. Staining was performed according to the manufacturer’s instructions. The stained sections were imaged using an Olympus BX43 microscope at different magnifications. The number of goblet cells in the colon (40× images) was statistically analyzed.

### 4.3. Immunohistochemistry

The immunohistochemistry protocol was the same as that reported previously [21]. Paraffin sections were processed accordingly and then incubated overnight with antibodies specific for LXRα (1:1000; home-made), LXRβ (1:1000; home-made), LXRαβ (1:100; Life Span Biosciences), MUC2 (1:2000; ab272692), TFF3 (1:5000; ab300427), Iba1 (1:1000; ab178846), Myeloperoxidase (1:500, ab45977), PCNA (1:8000; ab29), β-catenin (1:500; ab32572), α-catenin (1:100; sc7894), E-cadherin (1:200; sc7870), Plectin (1:100; BD Transduction), ChAT (1:1500; ab178850), ELOVL1 (1:200; LS-B13651), and ERβ (1:100; in-house antibody mapping the C-terminus part). Primary antibodies were replaced with BSA as negative controls. Sections were incubated with biotinylated secondary antibody (1:200) for 1 h at room temperature, and then the Vectastain Avidin-Biotin Complex kit (Vector Laboratories, Newark, CA, USA) was used according to the manufacturer’s instructions, followed by 3,3′-diaminobenzidine staining (Thermo Scientific, Waltham, MA, USA) and counter staining with Mayer’s hematoxylin (Sigma-Aldrich, St. Louis, MO, USA). Images were acquired and analyzed using an Olympus BX43 microscope.

### 4.4. Quantification of Immunostaining

Images were acquired with an Olympus BX43 microscope and saved as TIF files. Immunostaining results were examined and analyzed using ImageJ software [22]. Images were first grayscaled and then threshed to 8-bit black and white. The integrated density of plectin, ChAT, and ELOVL1 was quantified and normalized, respectively.

### 4.5. Data Analysis

Data are expressed as mean ± SD. The statistical significance of differences between groups was evaluated using Student’s *t*-test. *p* < 0.05 was considered to be statistically significant.

## 5. Conclusions

In conclusion, LXRs play an important role in regulating the function of the mouse colonic epithelium, and the inactivation of LXRαβ causes colitis similar to that seen in plectin knockout, but the direct regulator of the genes most changed in the LXRαβ^−/−^ mouse colon was ERβ.

## Figures and Tables

**Figure 1 ijms-24-12461-f001:**
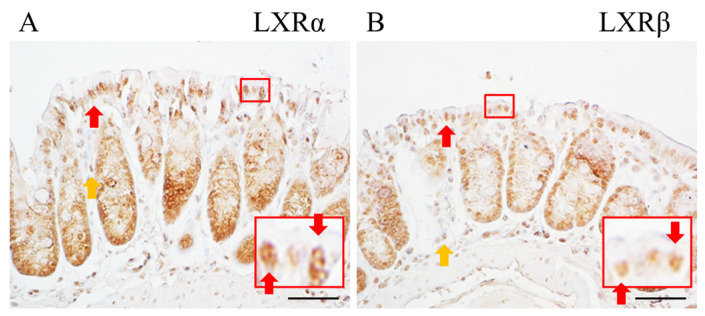
Expression of LXRα and LXRβ in mouse colon. In mouse colon, LXRα and LXRβ are expressed in the colonic epithelial cells ((**A**,**B**), respectively). Red arrows indicate positive epithelial cells and yellow arrows indicate other positive cells (macrophages). Scale bars: 50 μm.

**Figure 2 ijms-24-12461-f002:**
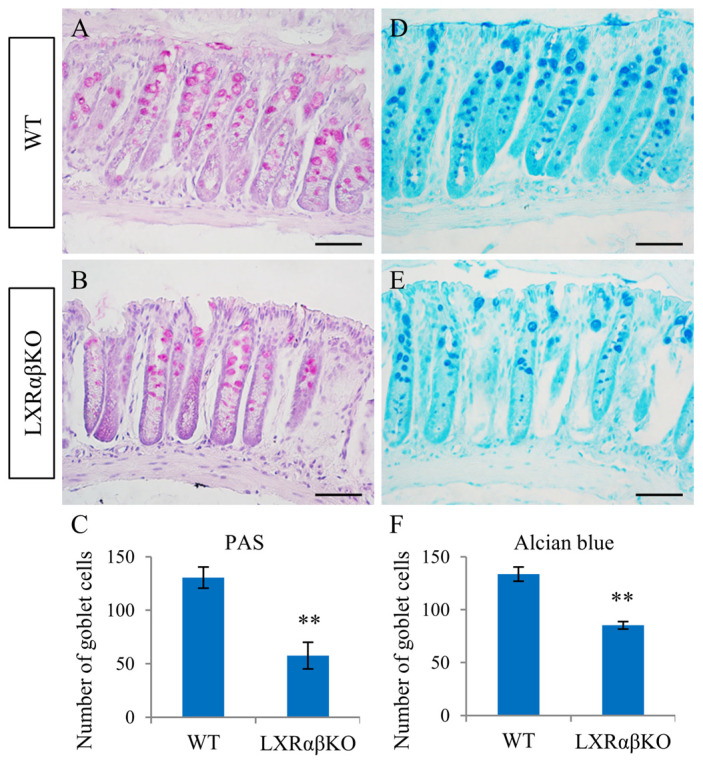
Decrease in the number of goblet cells in the colon of LXRαβ^−/−^ mice. Periodic acid–Schiff staining showed significant decrease (** *p* < 0.01) in the number of goblet cells in the colon of LXRαβ^−/−^ mice compared to WT (**A**–**C**). Alcian blue staining confirmed that goblet cells were significantly reduced (** *p* < 0.01) in LXRαβ^−/−^ mice (**D**–**F**). PAS: periodic acid–Schiff; *n* = 3. Scale bars (**A**,**B**,**D**,**E**): 50 μm.

**Figure 3 ijms-24-12461-f003:**
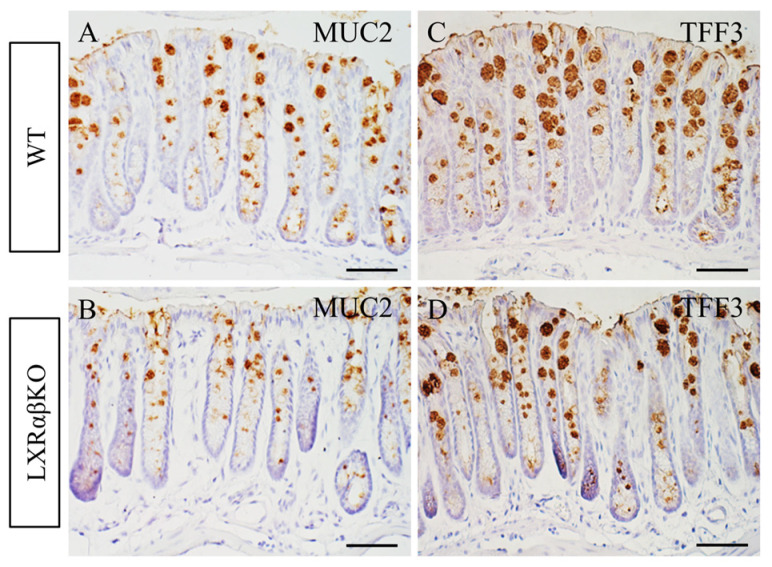
Decreased expression of MUC2 and TFF3 in the colon of LXRαβ^−/−^ mice. MUC2 was strongly expressed in the colon of WT mice (**A**), and its expression was decreased in the colon of LXRαβ^−/−^ mice (**B**). TFF3 was highly expressed in the colon of WT mice (**C**), and its expression was significantly decreased in the LXRαβ^−/−^ mice (**D**). Scale bars (**A**–**D**): 50 μm.

**Figure 4 ijms-24-12461-f004:**
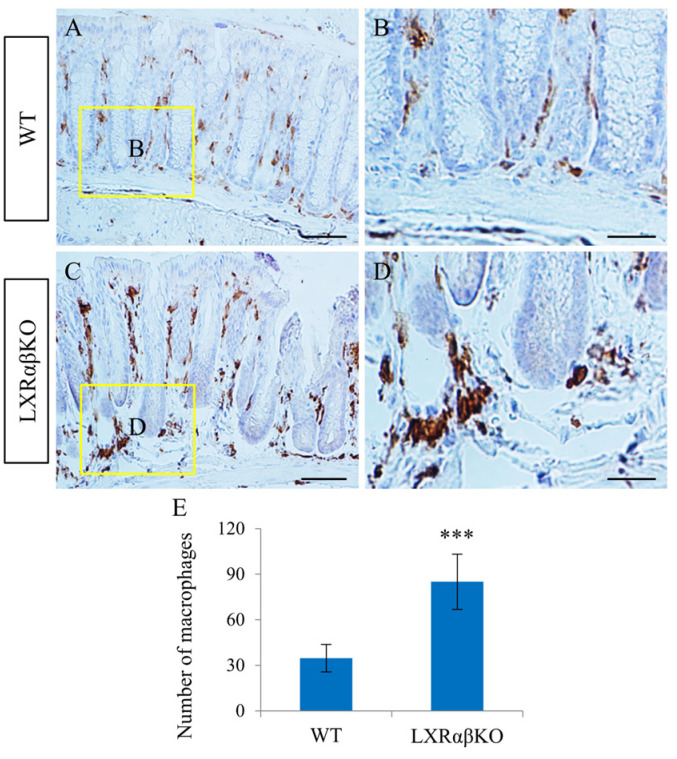
Increased macrophages in the lamina propria of LXRαβ^−/−^ mice. There were few Iba1-positive macrophages within the lamina propria of WT mice (**A**,**B**). In the colon of LXRαβ^−/−^ mice, the number of macrophages was much higher (*** *p* < 0.001) than that in WT and these macrophages appeared to be activated (**C**–**E**). *n* = 3. Scale bars: (**A**,**C**), 50 μm; (**B**,**D**), 20 μm.

**Figure 5 ijms-24-12461-f005:**
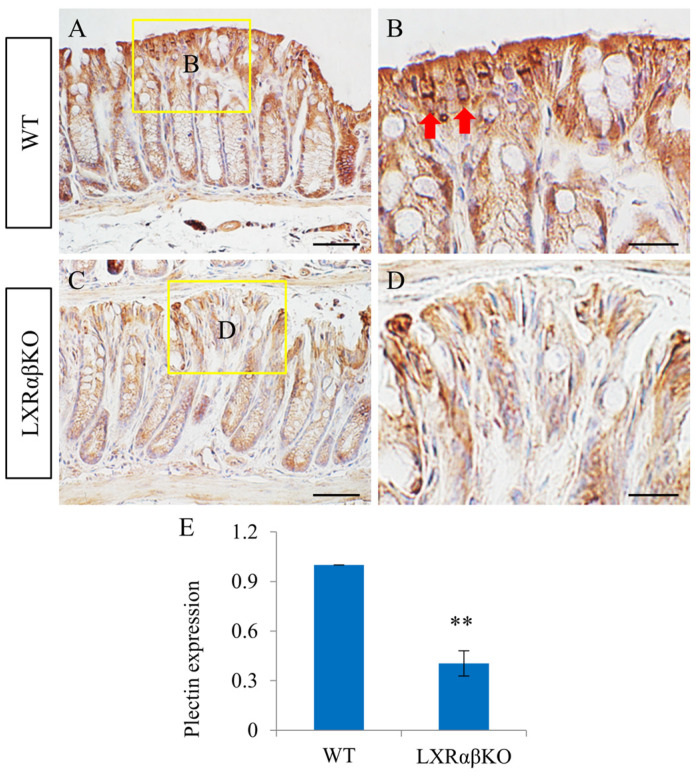
Decreased plectin expression in the colon of LXRαβ^−/−^ mice. There was a prominent expression of plectin in the cytoplasm and perinuclear regions of the colon in WT mice (**A**,**B**). Plectin expression in the colonic epithelium of LXRαβ^−/−^ mice was significantly decreased (** *p* < 0.01) (**C**–**E**). *n* = 3. Red arrows indicate the expression of plectin around the nucleus. Scale bars: (**A**,**C**), 50 μm; (**B**,**D**), 20 μm.

**Figure 6 ijms-24-12461-f006:**
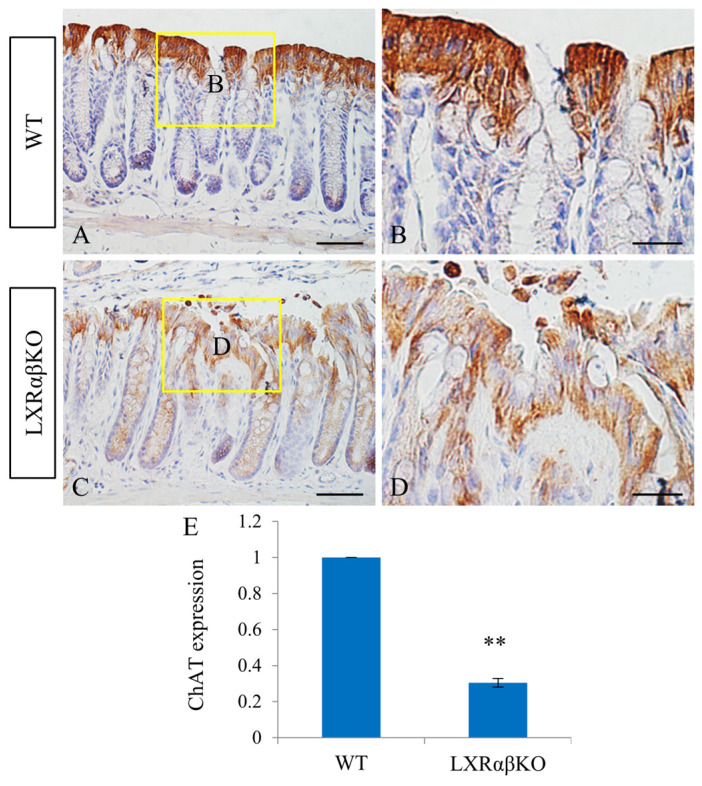
Decreased ChAT expression in the colonic epithelium of LXRαβ^−/−^ mice. There was abundant expression of ChAT in the colonic epithelium of WT mice (**A**,**B**). However, ChAT expression was markedly decreased (** *p* < 0.01) in the colonic epithelium of LXRαβ^−/−^ mice (**C**–**E**). *n* = 3. Scale bars: (**A**,**C**), 50 μm; (**B**,**D**), 20 μm.

**Figure 7 ijms-24-12461-f007:**
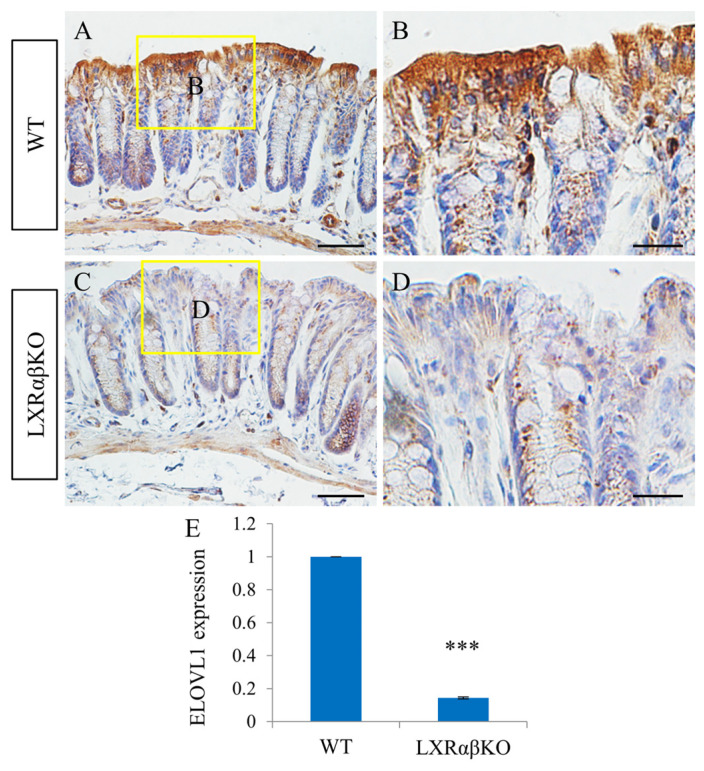
Decreased ELVOL1 expression in the colonic epithelium of LXRαβ^−/−^ mice. ELVOL1 was highly expressed in the colonic epithelium of WT mice (**A**,**B**). In the colonic epithelium of LXRαβ^−/−^ mice, ELVOL1 expression was significantly reduced (*** *p* < 0.001) (**C**–**E**). *n* = 3. Scale bars: (**A**,**C**), 50 μm; (**B**,**D**), 20 μm.

**Figure 8 ijms-24-12461-f008:**
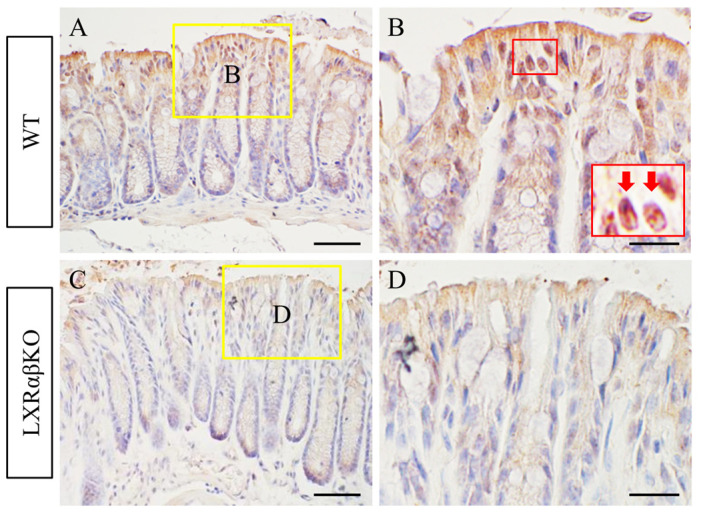
Loss of ERβ in the epithelium of LXRαβ^−/−^ mice. There was a very strong nuclear expression of ERβ in the colonic epithelium of WT mice (**A**,**B**). The expression of ERβ was significantly reduced in the colonic epithelium of LXRαβ^−/−^ mice (**C**,**D**). Red arrows indicate ERβ-positive cells. Scale bars: (**A**,**C**), 50 μm; (**B**,**D**), 20 μm.

## Data Availability

Data and materials used to support the findings of this study are available from the corresponding author upon request.

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
