# Peer review of "Loss of ERβ in Aging LXRαβ Knockout Mice Leads to Colitis"

_ijms, 2023, doi:10.3390/ijms241512461_

Round 1

Reviewer 1 Report

This manuscript by Xiaoyu Song et al. investigated the mechanisms through which loss of Liver X receptors (LXRs) leads to sensitivity of the colon to colitis. The authors conclude that colonic phenotype in LXRαβ-/- mice is not directly due to loss of LXRs, but was indirectly caused by the loss of ERβ in the colonic epithelium which occurs in the absence of LXRs. It is clear from study findings that ERβ expression was lost in the colonic epithelium of LXRαβ-/- mice. I have few suggestions to improve the manuscript.

-       Discuss in detail about loss of ERβ and development of colitis in LXRαβ-/- mice in the discussion section. Authors have a done a good job in summarizing their findings in the discussion section but failed to discuss in detail that loss of ERβ lead to colitis.

-       Explore the relationship between loss of Erβ and other findings noted in the study including decrease in goblet cells and mucins, increased activation of macrophages, decreased expression of plectin, ChAT, and ELVOL1.

Reviewer 2 Report

This is a well writen manuscript, however I have a few comments for the authors. Methods can can and should be described in greater detail. Also, in introduction, consider starting the second paragraph as the third paragraph. Finaly, in this case ethics committe approval is needed as the study involved sacrifice of animals but I will leave the final decision to the editors.
